# Inhibitory Effects of Selected Medicinal Plants on Bacterial Growth of Methicillin-Resistant *Staphylococcus aureus*

**DOI:** 10.3390/molecules27227780

**Published:** 2022-11-11

**Authors:** In-Geun Jung, Jae-Young Jeong, Seung-Hoon Yum, You-Jin Hwang

**Affiliations:** 1Department of Biomedical Engineering, College of Health Science, Gachon University, Incheon 21936, Korea; 2Department of Health Sciences & Technology, Gachon Advanced Institute for Health Sciences & Technology, Gachon University, Incheon 21999, Korea

**Keywords:** antibacterial, cytotoxicity, medicinal plants, multidrug-resistance, methicillin-resistant *Staphylococcus aureus*

## Abstract

Methicillin-resistant *Staphylococcus aureus* (MRSA) is a serious threat to global public health due to its capacity of tolerate conventional antibiotics. Medicinal plants are traditionally used to treat infectious diseases caused by bacterial pathogens. In the present study, 16 medicinal plants were screened for antibacterial activities to preselect more effective species. Ethanol extracts of selected medicinal plants (*Caesalpinia sappan* L., *Glycyrrhiza uralensis* Fisch., *Sanguisorba officinalis* L., and *Uncaria gambir* Roxb) were partitioned successively with different solvents (n-hexane, chloroform, ethyl acetate, 1-butanol, and water). Disc diffusion assay and broth microdilution were performed to evaluate the antibacterial activities of plant extracts and fractions against *Staphylococcus aureus* strains. Furthermore, the cytotoxicity of the extracts and fractions was determined against the human hepatoma (HepG2) and human lung carcinoma (A549) cell lines using a trypan blue exclusion method. A few extracts and fractions showed significant inhibitory effects on the bacterial growth of all tested strains, including multidrug-resistance (MDR) clinical isolates. The ethyl acetate fraction of *C. sappan* had the most potent effects with minimum inhibitory/bactericidal concentrations (MIC/MBC) of 31.2/62.5 μg/mL and showed low cytotoxicity with over 90% cell viability in both cells. Our results suggest that medicinal plants have considerable potential as alternatives to conventional antibiotics.

## 1. Introduction

Methicillin-resistant *Staphylococcus aureus* (MRSA) is a prevalent pathogen that causes severe infections in both the healthcare and community settings [1]. MRSA infections have emerged as a global public health burden, substantially increasing morbidity and mortality rates. In hospital settings, Asian countries have reported more MRSA prevalence than countries in Europe [2,3]. The prevalence of MRSA in some Asian countries is particularly high. For example, 73% of clinical isolates in South Korea were identified as MRSA according to reports of a regional resistance surveillance program in 2011 [4,5].

MRSA shows resistance to beta-lactam antibiotics such as penicillin, cephalosporin, and carbapenem and, over time, has become resistant to a few other antibiotic classes including fluoroquinolone, macrolide, aminoglycoside, and clindamycin [6,7]. MRSA infections are difficult to treat due to the capacity of tolerate conventional antibiotics [3]. Thus, new antibiotic agents with different mechanisms of action are urgently needed to control MRSA infections. In fact, the World Health Organization (WHO) has classified MRSA as a high-priority pathogen for the development of novel antibiotics [8].

To search for effective alternatives of antibiotics to treat bacterial infections, plant resources have shown great potential. The use of medicinal plants is still being considered a conventional treatment for diseases or syndromes worldwide [9,10]. Medicinal values of plants are derived from various bioactive compounds such as alkaloids, flavonoids, phenolic compounds, steroids, tannins, terpenoids, and other secondary metabolites [10]. Plant-derived compounds possess low-cost, less toxicity, fewer side effects, and lower propensity of resistance development [11,12]. Owing to these unique properties, medicinal plants are considered attractive alternatives to synthetic chemical antibiotics [13]. Screening of medicinal plants is a key to the research and development of potential antibiotic agents for therapeutic use [14].

Individual bioactive compounds in plant extracts might have little efficacy because they often act in combination with other compounds to control microbial infections [15]. The present study aimed to evaluate antibacterial activities of medicinal plants using plant extracts and fractions of extracts. A total of 16 medicinal plants were screened for antibacterial activities using a broth microdilution method. The inhibitory effects of extracts and fractions from selected medicinal plants on bacterial growth of *S. aureus* strains including MRSA clinical isolates were then investigated. To evaluate the cytotoxicity of extracts and fractions against HepG2 and A549 cell lines, we performed a cell viability assay.

## 2. Results and Discussion

In the present study, we confirmed the antibacterial activities of plant extracts and fractions from medicinal plants. To preselect the more effective species, antibacterial activities of ethanol extracts from 16 medicinal plants were evaluated with a broth microdilution method (Appendix A). Four medicinal plants were selected based on their inhibitory effects on bacterial growth of *S. aureus* strains. Information on these 16 species of medicinal plants is presented in Table 1.

Ethanol extracts of medicinal plants were prepared and partitioned according to the above process (Figure 1). Extraction yields using ethanol for 16 medicinal plants are shown in Table 1. The highest extraction yield was obtained from *S. baicalensis* (62.37%), followed by that from *U. gambir* (61.84%). Water and butanol fractions showed higher average yields at 15.40% and 9.95%, respectively, than ethyl acetate fractions (6.80%) and chloroform fractions (3.39%). Hexane fractions had the lowest average yield, at 0.56%. As the polarity of fraction solvents increased, more compounds or polar compounds with a higher mass were extracted from plants. These results are in line with previous studies showing that compounds in plant extracts are predominantly of intermediate and high polarity [16,17].

The inhibitory effects of selected medicinal plants were tested by antibacterial bioassays. Two reference strains (MSSA and MRSA) and two clinical isolates (MRSA and MDRSA) of *S. aureus* were used in these assays, showing different antibiotic resistance patterns (Table 2 and Appendix A). Thus, bioactive extracts with a novel mechanism of action could be confirmed. MDR was defined as being resistant to at least three different antibiotic classes [18].

Antibacterial activities of extracts and fractions were assessed by disc diffusion method. The diameter of a clear zone was measured and recorded. The results are shown in Table 3 and Table 4. All *S. aureus* strains were extremely susceptible to rifampicin. Distilled water exhibited no antibacterial activity. Antibacterial activities showed the following order from high to low: *C. sappan*, *U. gambir*, *G. uralensis*, and *S. officinalis* extracts (Appendix A). *C. sappan* extract showed a remarkable antibacterial activity with much wider inhibition zones (23.27–25.53 mm) at a concentration of 40 mg/mL and even at a lower concentration of 2.5 mg/mL. In contrast, *S. officinalis* extracts showed no antibacterial activities at concentrations below 20 mg/mL. We confirmed that the susceptibilities of each bacterial strain to different extracts of selected medicinal plants showed no significant differences (Figure 2). In other words, plant extracts might possess different mechanisms of action compared to conventional antibiotics. Thus, they could be active against bacterial strains regardless of their resistance [19,20].

Disc diffusion assays were also performed using fractions of selected medicinal plants at a concentration of 10 mg/mL. According to results presented in Appendix A, hexane and chloroform fractions only showed inhibition zones for *G. uralensis*. Ethyl acetate fractions showed more potent activities than other fractions. Butanol fractions showed antibacterial activities against *C. sappan* and *U. gambir*. In contrast, water fractions of all plant species and *S. officinalis* fractions showed no antibacterial activities. The ethyl acetate fraction of *C. sappan* resulted in the largest clear zones with diameters of 23.10–23.83 mm. The butanol fraction of *C. sappan* also showed significant activities with diameters of 16.40–17.77 mm for clear zones. These results are considered to be due to different classes of phytochemicals extracted by different solvents and plant materials. Compounds extracted from medicinal plants are mainly dependent on the type of solvent used in the extraction process. For example, polar solvents are used for extracting phenolic compounds, their glycosides, and saponins, whereas non-polar solvents are used to extract fatty acids and steroids [21,22]. On the other hand, although water fraction simultaneously had the highest yields, they showed poor inhibitory effects. These results suggest that massive amounts of compounds might be present in more polar fractions. However, not all compounds show antibacterial activities [23].

Antimicrobial mechanisms of plant extracts are not clearly understood yet. Meanwhile, we confirmed that inhibition zones of plant extracts were dependent on their concentrations and fraction solvents. Our results showed that ethanol extracts of *G. uralensis*, *S. officinalis*, and *U. gambir* had an equal MIC value of 250 μg/mL by the broth microdilution method (Table 5). However, differences in inhibition zones of extracts were observed (Table 3). This could be caused by difficulty in diffusion through agar of *G. uralensis* and *S. officinalis* extracts because the major active compounds of these extracts possessed high MWs. The diameter of the clear zone measured by the disc diffusion method is dependent on the degree of diffusion of the active substance. Previous studies have reported that the degree of diffusion is affected by the MW and polarity of compounds [23,24]. The above limitation of the disc diffusion method might have caused differences in the antimicrobial activity between different types of extracts. Disc diffusion method could be inappropriate for testing antimicrobial activities of plant extracts due to factors that might adversely affect the accuracy of results [24,25]. Thus, we determined MIC values to accurately compare the effectiveness of different plant extracts.

It is widely considered that extracts having MIC values below 8 mg/mL possess some antimicrobial activities, while extracts with MIC values below 1 mg/mL exhibit remarkable antimicrobial activities [7]. According to results presented in Appendix A, half of tested extracts showed remarkable antibacterial activities. Especially, four ethanol extracts (*C. sappan*, *G. uralensis*, *S. officinalis*, and *U. gambir*) exhibited more significant activities with lower MIC values (Table 5). The ethanol extract of *C. sappan* showed the best activity with MIC values of 62.5 μg/mL for all tested strains. Among various fractions, hexane, chloroform, and water fractions showed little activity. However, hexane and chloroform fractions of *G. uralensis* inhibited the bacterial growth of *S. aureus*, with MIC values of 62.5 and 31.2 μg/mL, respectively. MIC values of ethyl acetate fractions were determined as follows: *C. sappan* (31.2 μg/mL), *G. uralensis* (125 μg/mL), *S. officinalis* (500 μg/mL), and *U. gambir* (62.5 μg/mL). The MIC values of butanol fractions were *C. sappan* (62.5–125 μg/mL), *S. officinalis* (250 μg/mL), and *U. gambir* (125 μg/mL). The butanol and water fractions of *G. uralensis* showed little activity. A previous study has reported that the chloroform fraction of *G. uralensis* has 2.5-fold higher antimicrobial activity than the hexane fraction, with an MIC value of 0.1 mg/mL against MRSA strains, whereas the butanol and aqueous fractions exhibit no antibacterial activity [26]. Furthermore, prominent antibacterial activity was observed for extracts and fractions from *C. sappan*. Brazilin, one of the major active compounds in *C. sappan*, has been reported to show noteworthy antibacterial activity against antibiotic-resistant bacteria, including MRSA [27].

The MBC/MIC ratio determines whether extracts and fractions are bactericidal or bacteriostatic. The results are interpreted as bactericidal for MBC/MIC ≤ 4 or bacteriostatic for MBC/MIC > 4 [28]. The MBC/MIC ratio is shown in Table 5 and Figure 3. Ethanol extracts and water fractions of *S. officinalis* revealed bacteriostatic activities. Other extracts and fractions showed bactericidal activities. These results indicated that extracts and fractions of selected medicinal plants might act as potent bactericidal agents. Bactericidal agents are preferred over bacteriostatic agents clinically because killing bacteria is considered to be more effective for controlling infections than inhibiting bacterial growth [29].

To elucidate the inhibitory effects of extracts and fractions, a bacterial growth curve was determined toward MRSA 33,591 and CI-2. As shown in Figure 4, ethanol extracts of selected medicinal plants had significant bacterial growth inhibition with a wide range of concentrations. All tested bacteria showed dose-dependent growth inhibition. The time-lag with 31.2 μg/mL of *C. sappan* extract reaching to the exponential phase was changed from 8 h to 10 h compared to BHI broth (control). However, our results demonstrated that fractions showed much stronger inhibitory effects than ethanol extracts. The ethyl acetate fraction of *C. sappan* and chlorform fraction of *G. uralensis* extended the lag phase by 2 h and 4 h at a concentration of 15.6 μg/mL, respectively (Figure 5).

The cytotoxicity of extracts and fractions was determined using the trypan blue exclusion method. Trypan blue is a negatively charged dye and only stains non-viable cells in blue [30]. HepG2 and A549 cells were treated with MIC concentrations of extracts and fractions for 24 h. The number of viable cells was counted, and cell viability was calculated with the percentage of living cells and dead cells. As shown in Table 6, ethanol extract and ethyl acetate fraction of *C. sappan* had low cytotoxicity (>90% cell viability) at concentrations of 62.5 and 31.2 μg/mL, whereas other extracts and fractions showed mild cytotoxicity (60–90% cell viability) [31]. Cell viability increased at lower doses, and both cell lines showed a slight difference in the viability between different fraction solvents and plant materials but with no significant differences. These results suggest that plant extracts may be relative efficient and safe for therapeutic use. However, normal cell lines were not used in the present study and further studies are needed to demonstrate the safety of plant extracts. It is critically important to note that in vivo efficacy and toxicity of plant extracts may not reflect in vitro properties due to other pharmacokinetic and pharmacodynamic factors [32].

## 3. Materials and Methods

### 3.1. Plant materials

A total of 16 species of medicinal plants (*Areca catechu* L., *Caesalpinia sappan* L., *Cinnamomum loureirii* Nees., *Curcuma aromatica* Salisb., *Euphorbia humifusa* Wild., *Glechoma grandis* (A. Gray) Kuprian., *Glycyrrhiza uralensis* Fisch., *Lonicera japonica* Thunb, Morus alba L., *Phellinus linteus*, *Polygonum tinctorium* Ait., *Quercus salicina* Blume, *Sanguisorba officinalis* L., *Scutellaria baicalensis* Georgi., *Sophora flavescens* Ait., and *Uncaria gambir* Roxb.) were purchased from Samhong Medicinal Herb Market (Seoul, Korea).

### 3.2. Preparation and Fractionation of Plant Extracts

Plant materials were blended to powder using a home grinder and soaked in 70% ethanol with shaking (110 rpm) for 24 h. The ratio of plant materials to solvent was 1:10 (*w*/*v*). Crude extracts were centrifuged with 3000× *g* rpm for 30 min. Supernatants were collected and concentrated using a rotary evaporator WEV-1001V (Daihan Scientific Co., Wonju, Korea) in vacuum at 50 °C. The concentrated residue was dissolved in 10% dimethyl sulfoxide (DMSO; Sigma Chemical Co., St. Louis, MO, USA) and finally filtered through a Whatman filter paper No. 2 (Whatman, Kent, UK) to obtain ethanol extracts. 

For the preparation of fractions, ethanol extracts were suspended in distilled water and partitioned successively with different solvents (n-hexane, chloroform, ethyl acetate, and 1-butanol) (Sigma Chemical Co., St. Louis, MO, USA) in the order of increasing polarity. Each layer was prepared following the above process. All prepared samples were collected into conical tubes and stored in a refrigerator at 4 °C until further use. The percentage of extraction yield was calculated with the following equation: extraction yield (%) = (dry weight of extract/dry weight of plant material) × 100.

### 3.3. Bacteria Culture

Antibacterial activities of samples were evaluated against standard strains of *S. aureus* together with MRSA clinical isolates. Standard reference strains (*S. aureus* ATCC 29,213; MRSA ATCC 33,591) were purchased from American Type Culture Collection (ATCC; Manassas, VA, USA). MRSA clinical isolates were originally isolated from clinical specimens and identified in Gachon University Gil Medical Center (Incheon, Korea) [1]. These isolates were preserved in a −80 °C freezer in 20% glycerol (*v*/*v*) until further use. Each bacterium was initially cultivated on brain heart infusion (BHI; Kisan Bio, Seoul, Korea) agar medium in the plate. Before assays, a single colony was picked from each plate and incubated in BHI broth with shaking (110 rpm) at 37 °C for 24 h.

### 3.4. Antibiotic Susceptibility Testing

Antibiotic susceptibility was tested using the Kirby–Bauer disc diffusion method with some modifications [33]. Susceptibilities to ten different antibiotic discs (ampicillin (10 µg), penicillin G (10 IU), kanamycin (30 µg), gentamicin (10 µg), streptomycin (10 µg), tetracycline (30 µg), erythromycin (15 µg), vancomycin (30 µg), chloramphenicol (30 µg), and methicillin (5 µg) (Liofilchem, Teramo, Italy)) were evaluated. Each bacterial suspension was adjusted to a turbidity of McFarland 0.5 and inoculated to BHI plates. After antibiotic discs were carefully placed on BHI plates, plates were incubated at 37 °C for 24 h. The diameter of a clear zone was measured and interpreted according to the manufacturer’s instructions based on CLSI guidelines.

### 3.5. Disc Diffusion Assay

Disc diffusion assay was performed to evaluate antibacterial activities following a previous study [34]. Briefly, each bacterial suspension (1 × 10^8^ CFU/mL) was mixed well with 100 mL of fresh BHI medium containing 1.5% agar (*w*/*v*). The mixture consisting of bacterial suspension and BHI medium was poured onto Petri dishes. While solidifying the medium, 100 μL of each sample was loaded onto each paper-disc (8 mm/diameter). Rifampicin (30 µg) (Sigma Chemical Co., St. Louis, MO, USA) was used as a positive control. Loaded paper discs were gently placed onto plates. These plates were then incubated at 37 °C for 24 h. Diameters of clear zones were measured after incubation. The experiments were performed in triplicate.

### 3.6. Determination of Minimum Inhibitory Concentration (MIC)

Minimum inhibitory concentration (MIC) was determined using the broth microdilution method according to Bostanci et al. [35] with slight modifications. Briefly, 200 μL of the samples was inoculated to first wells of 96-well microplates and serially diluted two-fold (ranging from 2000 to 3.9 μg/mL). Then, 100 μL of each bacterial suspension (1 × 10^6^ CFU/mL) was inoculated to wells. The total volume was 200 μL per well. Rifampicin was used as a positive control. The optical density (OD) was measured at 595 nm with a spectrophotometer (Multiskan FC; Thermo Fisher Scientific, Waltham, MA, USA). Microplates were incubated continuously for up to 18 h for monitoring inhibitory effects of samples on bacterial growth. MIC was defined as the lowest concentration that inhibited visible growth of bacteria.

### 3.7. Determination of Minimum Bactericidal Concentration (MBC)

To determine minimum bactericidal concentration (MBC), the broth microdilution assay was used and 50 μL of the samples was taken from the well that inhibited visible growth of bacteria. Samples were transferred to a new microplate, and 150 μL of BHI broth was added to each well. Plates were incubated for 18 h. MBC was defined as the lowest concentration that killed 99.9% of bacteria.

### 3.8. Cell Culture

Human hepatoma, HepG2 (KCLB 88,065) and human lung carcinoma, A549 (KCLB 10,185) cell lines were purchased from Korean Cell Line Bank (KCLB, Seoul, Korea). These cells were grown in monolayers in 25 cm^2^ flasks with 3 mL of Roswell Park Memorial Institute Medium (RPMI 1640; Gibco, Grand Island, NY, USA) supplemented with 10% fetal bovine serum (FBS; Gibco, Grand Island, NY, USA) and 1% penicillin and streptomycin (Sigma Chemical Co., St. Louis, MO, USA). They were cultured at 37 °C in a CO_2_ cell chamber (MMM Group, Planegg, Germany) with 5% CO_2_. Upon reaching 80–90% confluence, sub-cultured cells were dissociated with a trypsin-ethylenediaminetetraacetic acid (Trypsin-EDTA; Gibco, Grand Island, NY, USA) solution. After centrifugation, the cells were re-suspended with culture medium to 1 × 10^6^ cells/mL and plated on 25 cm^2^ flasks. After 24 h incubation, logarithmic growth phase cells were used in the cell viability assay.

### 3.9. Cell Viability Assay

The cytotoxicity of samples was determined using the trypan blue exclusion method according to Ndlovu et al. [36] with slight modifications. The cells were seeded at a density of 5 × 10^4^ cells/well in 24-well microplates and grown to attachment overnight. Subsequently, the cells were exposed to samples at their MIC concentrations. Each sample was diluted with RPMI 1640 and added to each well. Microplates were incubated at 37 °C with 5% CO_2_ for 24 h. After incubation, trypan blue solution (Sigma Chemical Co., St. Louis, MO, USA) was mixed with trypsinized cell suspensions in a ratio of 1: 1. Ten microliters of the mixture were deposited on a hemocytometer. After counting cells, the percentage of living cells and dead cells was calculated. The experiments were performed in duplicate.

### 3.10. Statistical Analysis

Statistical analysis was performed using Microsoft Excel 2016 (Microsoft, Redmond, WA, USA) and SigmaPlot version 12.0 (Systat Software, San Jose, CA, USA). Average values were calculated as means and standard deviation (±SD). Statistical differences were assessed by analysis of variance (ANOVA). A *p*-value of less than 0.05 was considered statistically significant. 

## 4. Conclusions

In the present study, we confirmed that ethanol extracts and fractions of medicinal plants had potent antibacterial activities against *S. aureus* strains, including MRSA clinical isolates. Among the tested samples selected with abilities to inhibit of *S. aureus* strains, the ethyl acetate fraction of *C. sappan* was the most effective. The fraction showed remarkable inhibitory effects on bacterial growth with MIC/MBC values of 31.2/62.5 μg/mL and exhibited low cytotoxicity with more than 90% survival rates on HepG2 and A549 cell lines, respectively. The chloroform fraction of *G. uralensis* also showed significant bacterial growth inhibition. The present study demonstrated a considerable potential of medicinal plants as alternatives to conventional antibiotics. 

## Figures and Tables

**Figure 1 molecules-27-07780-f001:**
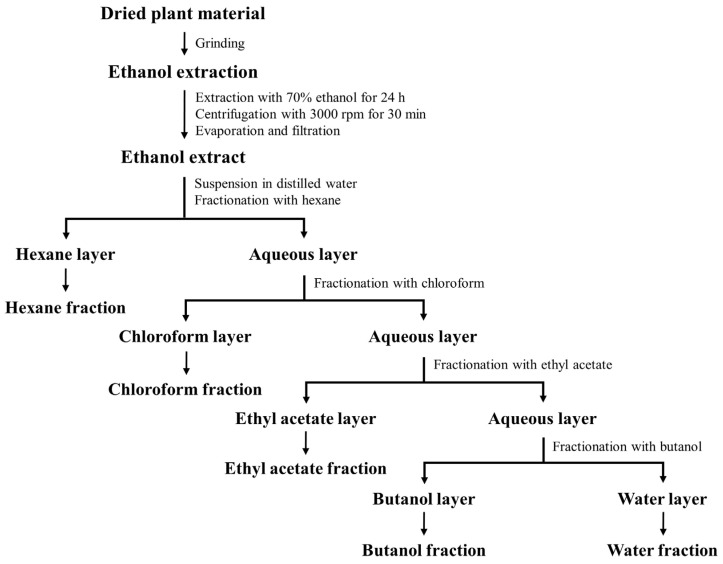
Extraction and fractionation scheme for preparation of ethanol extracts and fractions.

**Figure 2 molecules-27-07780-f002:**
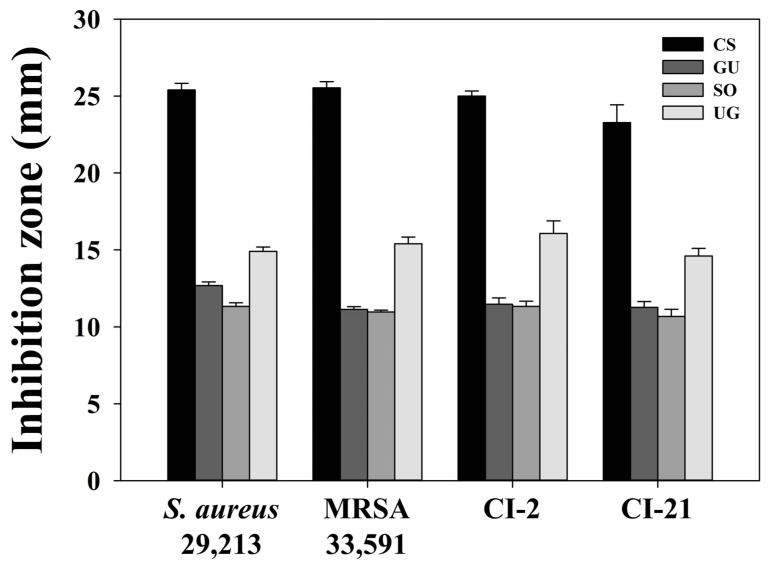
Inhibition zones of ethanol extracts (40 mg/mL) from selected medicinal plants against *S. aureus* strains. These results are represented the average diameter (mean ± SD) of triplicate tests, with *p* < 0.05 indicating statistical significance.

**Figure 3 molecules-27-07780-f003:**
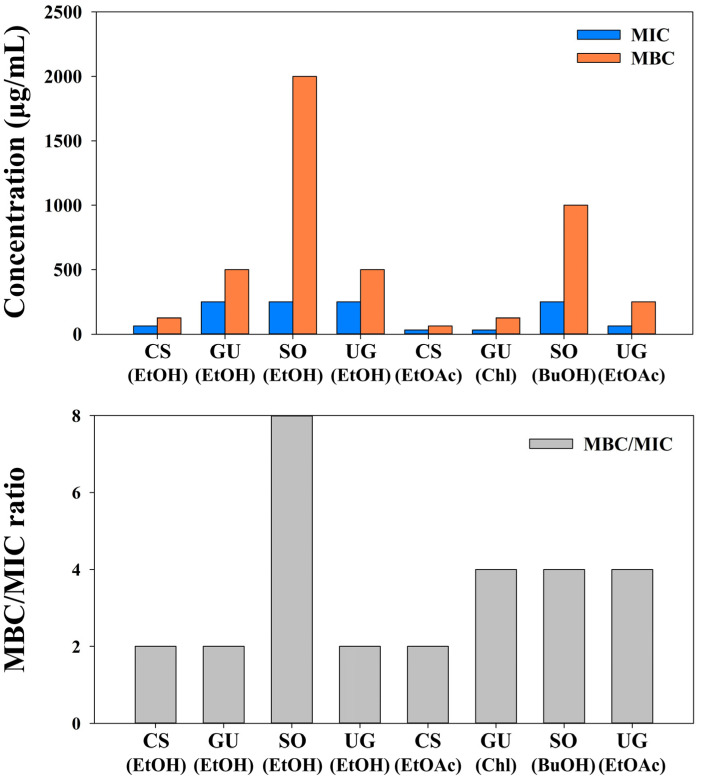
Antibacterial parameters of ethanol extracts and fractions from selected medicinal plants against MRSA strains.

**Figure 4 molecules-27-07780-f004:**
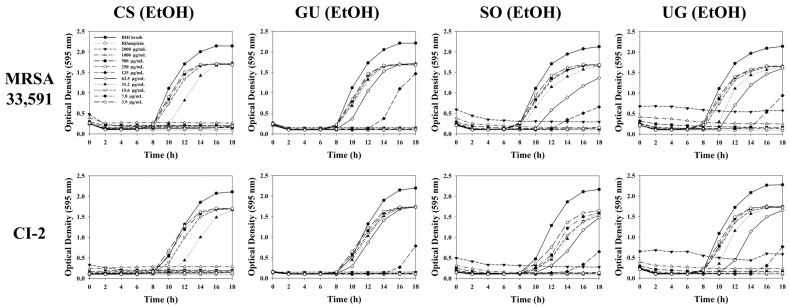
Influence of ethanol extracts from selected medicinal plants on bacterial growth of MRSA 33,591 and CI-2.

**Figure 5 molecules-27-07780-f005:**
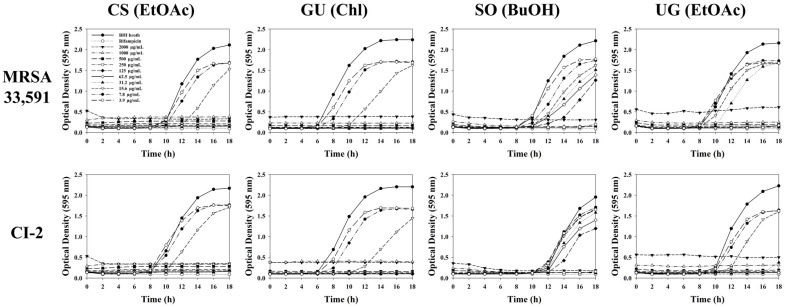
Influence of fractions from selected medicinal plants on bacterial growth of MRSA 33,591 and CI-2.

**Table 1 molecules-27-07780-t001:** List of medicinal plants used in the present study.

Scientific Name	Common Name	Family	Parts Used	Origin	Extraction Yield (%)
*Areca catechu* L.	Areca nut	Arecaceae	Seeds	Indonesia	17.62
*Caesalpinia sappan* L.	Sappan wood	Leguminosae	Heartwoods	Indonesia	10.47
*Curcuma aromatica* Salisb.	Wild turmeric	Zingiberaceae	Roots	Indonesia	19.65
*Cinnamomum loureiroi* Nees.	Saigon cinnamon	Lauraceae	Barks	Vietnam	11.48
*Euphorbia humifusa* Wild.	Creeping euphorbia	Euphorbiaceae	Leaves and stems	Korea	31.22
*Glechoma grandis* (A. Gray) Kuprian.	Ground ivy	Lamiaceae	Leaves and stems	Korea	35.16
*Glycyrrhiza uralensis* Fisch.	Chinese liquorice	Fabaceae	Roots	China	20.95
*Lonicera japonica* Thunb.	Japanese honeysucke	Caprifoliaceae	Floral buds	China	38.63
*Morus alba* L.	White mulberry	Moraceae	Leaves	Korea	20.25
*Phellinus linteus*	Sanghuang	Hymenochaetaceae	Fruit bodies	China	3.80
*Polygonum tinctorium* Ait.	Chinese indigo	Polygonaceae	Leaves	China	9.05
*Quercus salicina* Blume	Japanese willow leaf oak	Fagaceae	Leaves	Korea	31.92
*Sanguisorba officinalis* L.	Greater burnet	Rosaceae	Roots	China	23.84
*Scutellaria baicalensis* Georgi.	Baikal skullcap	Lamiaceae	Roots	Korea	62.37
*Sophora flavescens* Ait.	shrubby sophora	Leguminosae	Roots	Korea	29.70
*Uncaria gambir* Roxb.	Gambir	Rubiaceae	Leaves and twigs	Indonesia	61.84

**Table 2 molecules-27-07780-t002:** Antibiotic resistance profiles of *S. aureus* strains used in the present study.

Strains	Phenotype	Antibiotic Resistance Pattern
*S. aureus* ATCC 29,213	MSSA	-
MRSA ATCC 33,591	MRSA	Amp, Pen, Kan, Eryth, Strep, Tet, Gen, Chlo, Meth
CI-2 *	MDRSA	Amp, Pen, Kan, Eryth, Strep, Tet, Gen, Meth
CI-21 *	MRSA	Amp, Pen, Kan, Strep, Gen, Meth

*S. aureus*: *Staphylococcus aureus*; CI: Clinical isolate; MSSA: Methicillin-susceptible *Staphylococcus aureus*; MRSA: Methicillin-resistant *Staphylococcus aureus*; MDRSA: Multidrug-resistant *Staphylococcu aureus*; Amp: ampicillin; Pen: Penicillin G; Kan: Kanamycin; Eryth: Erythromycin; Gen: Gentamycin; Tet: Tetracyclin; Strep: Streptomycin; Van: Vancomycin; Chlo: Chloramphenicol; Meth: methicillin. * Clinical isolates were isolated and identified in the previous study [1].

**Table 3 molecules-27-07780-t003:** Antibacterial activities of ethanol extracts from selected medicinal plants against *S. aureus* strains.

Strains	Medicinal Plants	Diameter of the Clear Zone (mm)						
		Distilled Water	Rifampicin (30 μg)	Extract Concentration (mg/mL)				
				40	20	10	5	2.5
*S. aureus* 29,213	CS	-	29.87 ± 0.09	25.40 ± 0.43	20.67 ± 0.24	17.27 ± 0.52	12.43 ± 0.33	9.77 ± 0.21
	GU	-	29.87 ± 0.19	12.67 ± 0.24	10.00 ± 0.41	8.10 ± 0.08	-	-
	SO	-	30.00 ± 0.00	11.33 ± 0.24	8.80 ± 0.28	-	-	-
	UG	-	30.30 ± 0.51	14.90 ± 0.29	11.80 ± 0.24	9.17 ± 0.37	-	-
MRSA 33,591	CS	-	30.73 ± 0.57	25.53 ± 0.41	21.60 ± 0.43	18.07 ± 0.09	13.90 ± 0.65	10.73 ± 0.52
	GU	-	30.67 ± 0.47	11.13 ± 0.19	9.73 ± 0.57	8.17 ± 0.09	-	-
	SO	-	30.67 ± 0.47	10.97 ± 0.12	10.20 ± 0.16	-	-	-
	UG	-	30.67 ± 0.62	15.40 ± 0.43	12.33 ± 0.47	9.13 ± 0.12	-	-
CI-2	CS	-	30.00 ± 0.00	25.00 ± 0.33	21.93 ± 0.74	18.07 ± 0.09	13.87 ± 0.41	11.13 ± 0.34
	GU	-	29.67 ± 0.47	11.47 ± 0.41	9.60 ± 0.85	8.17 ± 0.08	-	-
	SO	-	29.67 ± 0.47	11.33 ± 0.34	9.87 ± 0.19	-	-	-
	UG	-	29.27 ± 0.90	16.07 ± 0.82	14.93 ± 0.82	10.00 ± 0.82	-	-
CI-21	CS	-	29.33 ± 0.47	23.27 ± 1.16	20.47 ± 0.34	17.50 ± 0.41	13.17 ± 0.24	9.87 ± 0.19
	GU	-	29.83 ± 0.62	11.27 ± 0.38	9.60 ± 0.85	8.20 ± 0.16	-	-
	SO	-	29.33 ± 0.47	10.67 ± 0.47	9.67 ± 0.47	-	-	-
	UG	-	29.27 ± 0.90	14.60 ± 0.49	12.27 ± 0.38	9.93 ± 0.09	-	-

*S. aureus*: *Staphylococcus aureus*; MRSA: Methicillin-resistant *Staphylococcus aureus*; CI: Clinical isolate; CS: *Caesalpinia sappan* L.; GU: *Glycyrrhiza uralensis* Fisch.; SO: *Sanguisorba officinalis* L.; UG: *Uncaria gambir* Roxb. These results are represented the average diameter (mean ± SD) of triplicate tests, with *p* < 0.05 indicating statistical significance.

**Table 4 molecules-27-07780-t004:** Antibacterial activities of fractions from selected medicinal plants against *S. aureus* strains.

Strains	Medicinal Plants	Diameter of the Clear Zone (mm)						
		Distilled Water	Rifampicin (30 μg)	Fraction Concentration (10 mg/mL)				
				HEX	Chl	EtOAc	BuOH	Water
*S. aureus* 29,213	CS	-	30.83 ± 0.24	-	-	23.23 ± 0.61	16.77 ± 0.95	-
	GU	-	30.60 ± 0.43	8.67 ± 0.47	11.93 ± 0.33	10.70 ± 0.22	-	-
	SO	-	30.67 ± 0.47	-	-	-	-	-
	UG	-	30.43 ± 0.42	-	-	14.67 ± 1.09	10.00 ± 0.00	-
MRSA 33,591	CS	-	31.50 ± 0.41	-	-	23.10 ± 1.22	16.77 ± 0.21	-
	GU	-	31.00 ± 0.00	8.33 ± 0.24	11.70 ± 0.79	10.93 ± 0.42	-	-
	SO	-	30.67 ± 0.47	-	-	-	-	-
	UG	-	30.83 ± 0.24	-	-	16.33 ± 1.09	9.93 ± 0.09	-
CI-2	CS	-	31.67 ± 0.24	-	-	23.83 ± 1.10	17.77 ± 0.92	-
	GU	-	30.83 ± 0.24	8.67 ± 0.47	11.37 ± 0.49	10.57 ± 0.49	-	-
	SO	-	30.93 ± 0.09	-	-	-	-	-
	UG	-	31.43 ± 0.33	-	-	14.43 ± 0.66	10.27 ± 0.21	-
CI-21	CS	-	31.83 ± 0.24	-	-	23.33 ± 0.50	16.40 ± 0.83	-
	GU	-	30.90 ± 0.29	8.77 ± 0.21	10.93 ± 0.09	10.23 ± 0.21	-	-
	SO	-	31.00 ± 0.00	-	-	-	-	-
	UG	-	31.33 ± 0.24	-	-	14.83 ± 1.68	9.83 ± 0.29	-

*S. aureus*: *Staphylococcus aureus*; MRSA: Methicillin-resistant *Staphylococcus aureus*; CI: Clinical isolate; CS: *Caesalpinia sappan* L.; GU: *Glycyrrhiza uralensis* Fisch.; SO: *Sanguisorba officinalis* L.; UG: *Uncaria gambir* Roxb.; HEX: n-Hexane; Chl: Chloroform; EtOAc: Ethyl acetate; BuOH: 1-Butanol; Water: Distilled water. These results are represented the average diameter (mean ± SD) of triplicate tests, with *p* < 0.05 indicating statistical significance.

**Table 5 molecules-27-07780-t005:** Minimum inhibitory concentration (MIC) and minimum bactericidal concentration (MBC) of ethanol extracts and fractions from selected medicinal plants against *S. aureus* strains.

Medicinal Plants	Extracts	Fractions	*S. aureus* 29,213			MRSA 33,591			CI-2			CI-21		
			MIC	MBC	MBC/MIC	MIC	MBC	MBC/MIC	MIC	MBC	MBC/MIC	MIC	MBC	MBC/MIC
CS	EtOH		62.5	125	2	62.5	125	2	62.5	125	2	62.5	125	2
		HEX	2000	>2000	ND	2000	>2000	ND	2000	>2000	ND	2000	>2000	ND
		Chl	1000	2000	2	1000	2000	2	1000	2000	2	1000	1000	1
		EtOAc	31.2	62.5	2	31.2	62.5	2	31.2	62.5	2	31.2	62.5	2
		BuOH	62.5	125	2	125	125	1	125	125	1	125	125	1
		Water	1000	2000	2	1000	2000	2	1000	2000	2	1000	2000	2
GU	EtOH		250	500	2	250	500	2	250	500	2	250	500	2
		HEX	62.5	250	4	62.5	250	4	62.5	250	4	62.5	250	4
		Chl	31.2	125	4	31.2	125	4	31.2	125	4	31.2	125	4
		EtOAc	125	500	4	125	500	4	125	500	4	125	500	4
		BuOH	>2000	ND	ND	>2000	ND	ND	>2000	ND	ND	>2000	ND	ND
		Water	>2000	ND	ND	>2000	ND	ND	>2000	ND	ND	>2000	ND	ND
SO	EtOH		250	2000	>4	250	2000	>4	250	2000	>4	250	2000	>4
		HEX	1000	>2000	ND	1000	>2000	ND	1000	>2000	ND	1000	>2000	ND
		Chl	1000	>2000	ND	1000	>2000	ND	1000	>2000	ND	1000	>2000	ND
		EtOAc	500	1000	2	500	2000	4	500	2000	4	500	2000	4
		BuOH	250	1000	4	250	1000	4	250	1000	4	250	1000	4
		Water	250	2000	>4	250	2000	>4	250	2000	>4	250	2000	>4
UG	EtOH		250	500	2	250	500	2	250	500	2	250	500	2
		HEX	>2000	ND	ND	>2000	ND	ND	>2000	ND	ND	>2000	ND	ND
		Chl	1000	1000	1	1000	2000	2	1000	2000	2	1000	2000	2
		EtOAc	62.5	125	2	62.5	250	4	62.5	250	4	62.5	250	4
		BuOH	125	250	2	125	250	2	125	250	2	125	250	2
		Water	250	1000	4	250	1000	4	250	1000	4	250	1000	4

CS: *Caesalpinia sappan* L.; GU: *Glycyrrhiza uralensis* Fisch.; SO: *Sanguisorba officinalis* L.; UG: *Uncaria gambir* Roxb.; EtOH: Ethanol; HEX: n-Hexane; Chl: Chloroform; EtOAc: Ethyl acetate; BuOH: 1-Butanol; Water: Distilled water; *S. aureus*: *Staphylococcus aureus*; MRSA: Methicillin-resistant *Staphylococcus aureus*; CI: Clinical isolate; MIC: Minimum inhibitory concentration; MBC: Minimum bactericidal concentration; ND: No data. MIC and MBC values are presented as μg/mL.

**Table 6 molecules-27-07780-t006:** Cytotoxicity of ethanol extracts and fractions from selected medicinal plants on HepG2 and A549 cell lines.

Medicinal Plants	Extracts/Fractions	Cell Viability (%)	
		**HepG2**	**A549**
CS	EtOH	90.15 ± 0.48	91.94 ± 1.24
	EtOAc	90.75 ± 0.43	90.99 ± 0.50
GU	EtOH	81.24 ± 0.58	83.05 ± 0.28
	Chl	85.38 ± 2.62	87.94 ± 0.44
SO	EtOH	81.30 ± 0.52	81.92 ± 1.07
	BuOH	79.71 ± 0.30	80.64 ± 0.21
UG	EtOH	82.05 ± 0.56	83.43 ± 0.57
	EtOAc	85.45 ± 0.26	85.14 ± 0.29

CS: *Caesalpinia sappan* L.; GU: *Glycyrrhiza uralensis* Fisch.; SO: *Sanguisorba officinalis* L.; UG: *Uncaria gambir* Roxb.; EtOH: Ethanol; Chl: Chloroform; EtOAc: Ethyl acetate; BuOH: 1-Butanol.

## Data Availability

Data supporting this study are available on request from the corresponding author.

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
