# Peer review of "Inhibitory Effects of Selected Medicinal Plants on Bacterial Growth of Methicillin-Resistant Staphylococcus aureus"

_molecules, 2022, doi:10.3390/molecules27227780_

Round 1
Reviewer 1 Report
All findings are interesting, and the article includes a balanced and critical view of the outcomes. Following comments can improve the quality of articles
1. In my opinion, the research background should be improved. As mentioned in the Introduction section, To search for effective alternatives of antibiotics to treat bacterial infections, plant resources have shown great potential. The use of medicinal plants is still being considered a conventional treatment for diseases or syndromes worldwide. However, this important claim was not supported by any documented references. In order to balance the scientific viewpoint and attract more attention from audiences, the authors are highly recommended to consider the inclusion of the some recent relevant research and review papers like (https://doi.org/10.3390/molecules27185847; ttps://doi.org/10.3390/antibiotics10030231; https://doi.org/10.3390/antibiotics9060346 ).
2. What is selection criteria for medicinal plant species used in this study
3. The graphical representation of antibacterial activity along with strong statistical analysis for fair scientific comparison in R, Origin, SPSS or any statistical software is highly recommended
4. What is selection criteria for reference strains and clinical isolates?
5. Graphically present the comparison of antibacterial activity of different strains
6. Graphically present the comparison of minimum inhibitory concentration (MIC) and bactericidal concentrations (MBS).
7. The quality of Figure 2 and 3 is poor. Need to improve
8. How you selected the test concentration for antibacterial activity
9. Conclusions need to be revised as per own results
10. The whole article needs to be checked critically for typos and grammar errors
11. All the references should be according to Journal guidelines

Author Response
Title: Inhibitory effects of selected medicinal plants on bacterial growth of methicillin-resistant Staphylococcus aureus
General Comments: Thank you for the opportunity to address the reviewer's constructive comments and revise the manuscript accordingly. We also appreciate the time and effort that
reviewers dedicated to providing the comments. All revisions in the manuscript are marked up using the “Track Changes” function.
Reviewer 1
Thank you for the opportunity to review this manuscript, dealing with interesting findings entitled “Inhibitory effects of selected medicinal plants on bacterial growth of methicillinresistant Staphylococcus aureus”. All findings are interesting, and the article includes abalanced and critical view of the outcomes. Following comments can improve the quality of articles
- In my opinion, the research background should be improved. As mentioned in the Introduction section, To search for effective alternatives of antibiotics to treat bacterial infections, plant resources have shown great potential. The use of medicinal plants is still being considered a conventional treatment for diseases or syndromes worldwide. However, this important claim was not supported by any documented references. In order to balance the scientific viewpoint and attract more attention from audiences, the authors are highly recommended to consider the inclusion of the some recent relevant research and review papers like (https://doi.org/10.3390/molecules27185847; https://doi.org/10.3390/antibiotics10030231; https://doi.org/10.3390/antibiotics9060346 ).
Response #1. Thank you for pointing out. We have included the review paper and modified all of reference numbering.
Reference no. 9. Alibi, S.; Crespo, D.; Navas, J., Plant-Derivatives Small Molecules with Antibacterial Activity. Antibiotics (Basel) 2021, 10, (3).
- What is selection criteria for medicinal plant species used in this study
Response #2. We have selected 16 species of medicinal plants used in traditional Asian medicine for the treatment of various diseases including microbial infections.
- 3. The graphical representation of antibacterial activity along with strong statistical analysis for fair scientific comparison in R, Origin, SPSS or any statistical software is highly recommended
Response #3. Thank you for the suggestion. We have presented it.
- 4. What is selection criteria for reference strains and clinical isolates?
Response #4. We have selected two reference strains (MSSA and MRSA) and two clinical isolates (MRSA and MDRSA). Reference strains with different susceptibility to methicillin were selected as controls. Clinical isolates were selected to have different resistance patterns (Reference No 1: Mun & Hwang, Antibiotics 2019, No 3: Li et al., Biocell 2019).
- 5. Graphically present the comparison of antibacterial activity of different strains
Response #5. Thank you for the suggestion. We have presented in Figure 2.
- 6. Graphically present the comparison of minimum inhibitory concentration (MIC) and bactericidal concentrations (MBC).
Response #6. Thank you for the suggestion. We have presented in Figure 3.
- 7. The quality of Figure 2 and 3 is poor. Need to improve
Response #7. Thank you for pointing out. We have modified it (Figure 4 and 5).
- 8. How you selected the test concentration for antibacterial activity
Response #8. We performed preliminary experiments and referred to previous studies to select the test concentration for antibacterial activity.
- 9. Conclusions need to be revised as per own results
Response #9. Thank you for pointing out. We have revised conclusion.
- 10. The whole article needs to be checked critically for typos and grammar errors
Response #10. Thank you for pointing out. We checked and have modified it.
- 11. All the references should be according to Journal guidelines
Response #11. Thank you for pointing out. We checked and have modified it.

Reviewer 2 Report
In this manuscript, authors present the antibacterial properties of extracts of several medicinal plants regarding strains of methicillin-resistant S. aureus. In addition, the cytotoxic effect of these extracts is studied by examining the effect on the viability of two cell lines.
In literature, many of these plants have been studied in terms of their bactericidal action on many pathogens, including S. aureus. Nevertheless, the study is well organised and the results are presented clearly.
minor points:
I would recommend the use of the term 'susceptible" instead of "sensitive" eg line 103 "susceptible to rifampicin" instead of "sensitive to rifampicin."
Author Response
Reviewer 2
In this manuscript, authors present the antibacterial properties of extracts of several medicinal plants regarding strains of methicillin-resistant S. aureus. In addition, the cytotoxic effect of these extracts is studied by examining the effect on the viability of two cell lines. In literature, many of these plants have been studied in terms of their bactericidal action on many pathogens, including S. aureus. Nevertheless, the study is well organised and the results are presented clearly.
minor points:
I would recommend the use of the term 'susceptible" instead of "sensitive" eg line 103 "susceptible to rifampicin" instead of "sensitive to rifampicin."
Response #1. Thank you for the suggestion. We have changed word.
Line 90: “methicillin-sensitive” to “methicillin-susceptible”
Line 109: “sensitive to rifampicin” to “susceptible to rifampicin”
Line 115: “sensitivities of each bacterial strain” to “susceptibilities of each bacterial strain”

Reviewer 3 Report
The experiment was a simple straight forward study to show the antibacterial activity of plan extracts.
There is only 1 comment:
Only human cancer cells were used as control but normal cell lines were not used. It was difficult to demonstrate the safety of the extracts.
Author Response
Reviewer 3
The experiment was a simple straight forward study to show the antibacterial activity of plant extracts.
There is only 1 comment:
Only human cancer cells were used as control but normal cell lines were not used. It was difficult to demonstrate the safety of the extracts.
Response #1. Thank you for pointing out. We agree with the reviewer's comment and this is a limitation of the present study. We have added results and discussion.
Line 225-227: “However, normal cell lines were not used in the present study and further studies are needed to demonstrate the safety of plant extracts.
